# Inhibition of cholesterol biosynthesis through RNF145-dependent ubiquitination of SCAP

Li Zhang[1], Prashant Rajbhandari[1], Christina Priest[1], Jaspreet Sandhu[1], Xiaohui Wu[2], Ryan Temel[3,4], Antonio Castrillo[5,6], Thomas Q de Aguiar Vallim[2], Tamer Sallam[2], Peter Tontonoz[1]*

[1]Department of Pathology and Laboratory Medicine, Howard Hughes Medical Institute, University of California, Los Angeles, Los Angeles, United States; [2]Department of Medicine, Division of Cardiology, University of California, Los Angeles, Los Angeles, United States; [3]Saha Cardiovascular Research Center, University of Kentucky, Lexington, United States; [4]Department of Pharmacology and Nutritional Sciences, University of Kentucky, Lexington, United States; [5]Instituto de Investigaciones Biomédicas Alberto Sols, CSIC-Universidad Autónoma de Madrid, Unidad de Biomedicina-Universidad de Las Palmas de Gran Canaria (Unidad asociada al CSIC), Las Palmas de Gran Canaria, Spain; [6]Instituto Universitario de Investigaciones Biomédicas y Sanitarias, Universidad de Las Palmas de Gran Canaria, Las Palmas de Gran Canaria, Spain

**Abstract** Cholesterol homeostasis is maintained through concerted action of the SREBPs and LXRs. Here, we report that RNF145, a previously uncharacterized ER membrane ubiquitin ligase, participates in crosstalk between these critical signaling pathways. RNF145 expression is induced in response to LXR activation and high-cholesterol diet feeding. Transduction of RNF145 into mouse liver inhibits the expression of genes involved in cholesterol biosynthesis and reduces plasma cholesterol levels. Conversely, acute suppression of RNF145 via shRNA-mediated knockdown, or chronic inactivation of RNF145 by genetic deletion, potentiates the expression of cholesterol biosynthetic genes and increases cholesterol levels both in liver and plasma. Mechanistic studies show that RNF145 triggers ubiquitination of SCAP on lysine residues within a cytoplasmic loop essential for COPII binding, potentially inhibiting its transport to Golgi and subsequent processing of SREBP-2. These findings define an additional mechanism linking hepatic sterol levels to the reciprocal actions of the SREBP-2 and LXR pathways.
DOI: https://doi.org/10.7554/eLife.28766.001

*For correspondence:
ptontonoz@mednet.ucla.edu

## Introduction

Precise control of cholesterol levels is critical for the normal function of cells and organisms. The maintenance of cholesterol homeostasis relies on a delicate balance between production, uptake and output. Two transcription factor families with opposing actions – the liver-X receptors (LXRs) and the sterol regulatory element-binding proteins (SREBPs) – are the main orchestrators of this balance (*Hong and Tontonoz, 2014*; *Horton et al., 2002*).

LXRs are nuclear receptors activated by cholesterol metabolite oxysterols in the cell (*Fu et al., 2001*; *Janowski et al., 1996*; *Lehmann et al., 1997*). There are two LXR receptors, LXRα and LXRβ, that share a high degree of functional overlap (*Castrillo and Tontonoz, 2004*). In response to elevated cellular cholesterol levels, they function to induce the expression of genes involved in the

efflux of cholesterol, including *Abca1* and *Abcg1* and *Apoe* (*Kennedy et al., 2001*; *Laffitte et al., 2001*; *Repa et al., 2000b*; *Venkateswaran et al., 2000*). Abrogation of LXR expression in mouse models results in cholesterol accumulation and accelerated atherosclerosis (*Peet et al., 1998*; *Schuster et al., 2002*). Pharmacological activation of LXRs with synthetic agonists in mice confers protection against atherosclerosis (*Joseph et al., 2002*; *Terasaka et al., 2003*).

On the other hand, SREBPs control the transcription of lipid biosynthesis genes. SREBP-2 preferentially regulates the expression of genes involved in cholesterol production (*Horton et al., 2003*). Immature SREBP-2 protein is retained in the ER through the interaction with SCAP and INSIG proteins. In response to low cellular cholesterol levels, SCAP undergoes a conformational change that permits it to move from ER to Golgi in COPII vesicles, where it is processed to its mature form. The mature SREBP-2 transcription factor then translocates to the nucleus where it binds to its target promoters (*Goldstein et al., 2006*). Deletion of *Srebf2* in mice is embryonically lethal (*Shimano et al., 1997*; *Vergnes et al., 2016*). *Scap* knockout mice show reduced nuclear SREBP accumulation and low cholesterol levels in serum and liver (*Matsuda et al., 2001*).

Given the importance of the LXR and SREBP pathways in lipid homeostasis it is not surprising that mechanisms for crosstalk between them have emerged. For example, LXR responds to elevated hepatic sterol levels by stimulating fatty acid synthesis through transcriptional induction of *Srebp1c* expression (*Repa et al., 2000a*). SREBP-2 acts as a positive regulator of *Abca1* gene expression by enabling the generation of oxysterol ligands for LXR (*Wong et al., 2006*). Furthermore, an intronic microRNA located within *Srebf2* gene, *miR-33*, inhibits the expression of *Abca1* (*Najafi-Shoushtari et al., 2010*; *Rayner et al., 2010*). LXR inhibits cellular cholesterol uptake in an SREBP-independent fashion through the induction of the E3 ubiquitin ligase IDOL (*Zelcer et al., 2009*). LXR was also recently shown to modulate cholesterol biosynthesis in mouse liver by inducing the expression of the non-coding RNA *LeXis* (*Sallam et al., 2016*).

Here, we describe an addition mechanism for LXR-SREBP crosstalk. We identify RNF145, an ER membrane ubiquitin E3 ligase with no previously known function, as an inhibitor of hepatic SREBP-2 processing. Expression of *Rnf145* is induced in response to LXR activation and high-cholesterol diet feeding. Overexpression of *Rnf145* inhibits the expression of cholesterologenic genes, whereas inactivation of *Rnf145* stimulates it. We show that RNF145 induces the ubiquitination of SCAP on two lysine residues that lie in proximity to COP-II binding site, thereby inhibiting its transport to Golgi and the subsequent processing of SREBP-2. These findings define an additional mechanism linking hepatic sterol levels to the reciprocal actions of the SREBP and LXR pathways.

## Results

### RNF145 is an LXR target gene encoding an ER-localized E3 ubiquitin ligase

Inspired by our prior work on the E3 ubiquitin ligase IDOL and its role in mediating effects of the LXR pathway on lipoprotein receptor levels (*Zelcer et al., 2009*), we endeavored to identify addition E3 ligases whose expression was responsive to LXRs. The gene encoding the putative E3 ligase RNF145 was previously identified as a potential target gene of LXR by RNA-seq analysis in a genome-wide transcriptional profiling of primary mouse hepatocytes treated with the synthetic LXR ligand, GW3965 (*Sallam et al., 2016*) and this finding was recently confirmed by another group (*Cook et al., 2017*). To further investigate the regulation of *Rnf145* by LXR, we treated primary mouse hepatocytes from wild-type (WT) mice or those lacking expression of both LXRα and LXRβ mice (LXR DKO) with GW3965 alone, or in combination with the RXR ligand LG268. *Rnf145* expression in primary hepatocytes from WT mice was increased two fold with GW3965 treatment (*Figure 1A*). The combination of LXR and RXR ligands did not further induce *Rnf145* expression. In contrast, the stimulation of *Rnf145* expression by LXR ligand was lost in primary hepatocytes from LXR DKO mice (*Figure 1A*), indicating that it was dependent on the expression of LXRs.

We proceeded to investigate the regulation of *Rnf145* expression by LXR in vivo. Administration of GW3965 to both mice and non-human primates induced the expression of *Rnf145* in liver (*Figure 1B and C*). Moreover, feeding a western diet high in cholesterol prominently induced hepatic expression of *Rnf14*5 in WT C57BL/6 but not *Lxrαβ−/−* mice (*Figure 1D*), indicating that hepatic expression of *Rnf145* was responsive to dietary cholesterol in an LXR-dependent manner.

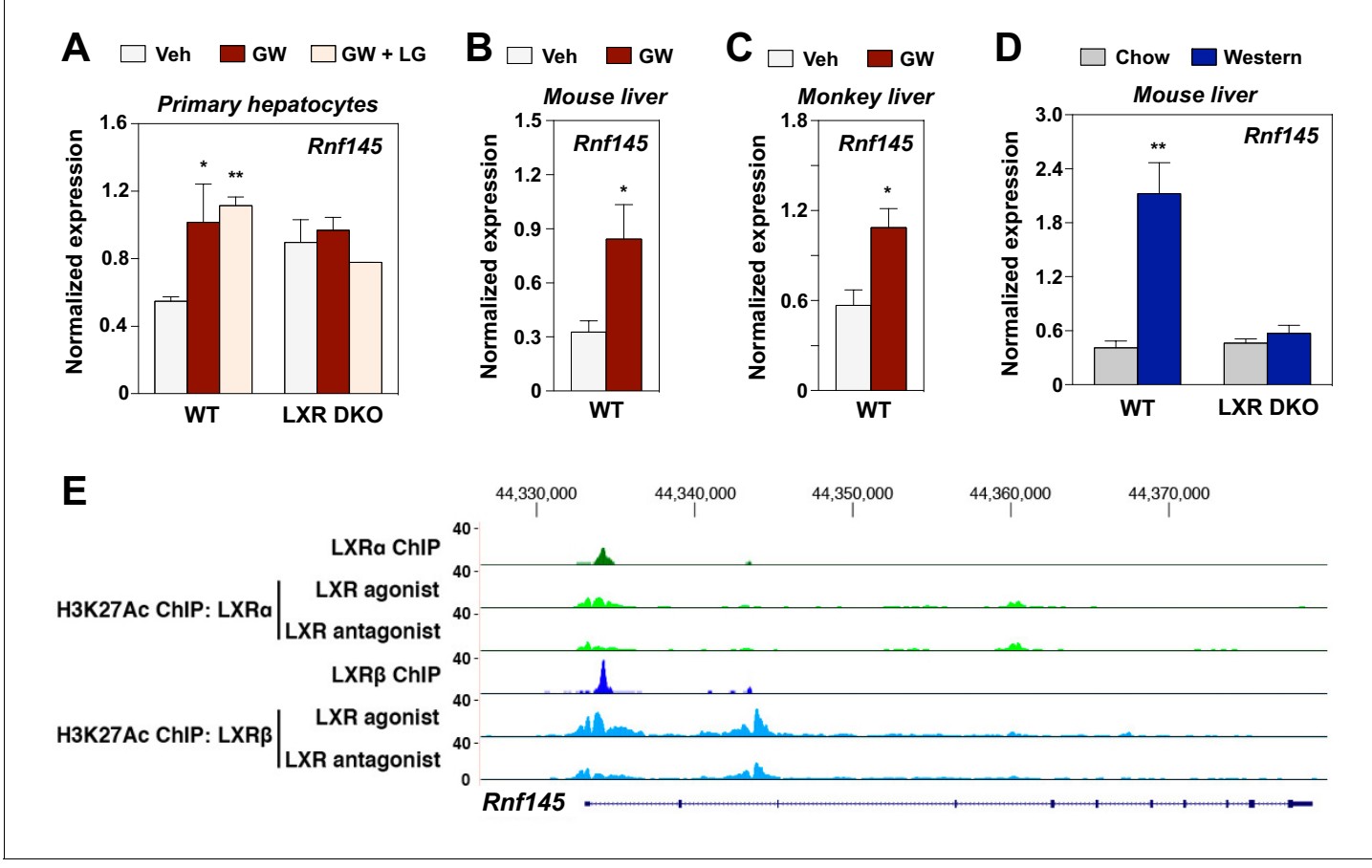

**Figure 1.** RNF145 is an LXR-regulated ubiquitin E3 ligase gene. (A) Expression levels of *Rnf145* as determined by real-time PCR analysis in primary hepatocytes from wild type or LXR DKO mice treated overnight with DMSO control, GW3965 (1 µM) or the combination of GW3965 (1 µM) and LG268 (50 nM). Results were presented as mean ± standard deviation (SD). (B) Expression levels of *Rnf145* in the liver tissue of mice orally gavaged with GW3965 (40 mg/kg/day) for 3 days. N = 5 per group. (C) Expression levels of *RNF145* in the liver tissue of cynomolgus monkeys orally gavaged with GW3965 (10 mg/kg/day) for 2 days. N = 5 per group. (D) Expression levels of *Rnf145* in the liver tissue of wild-type or LXR DKO mice placed on Western diet for 2 weeks. N = 8 per group. (E) ChIP-seq analysis showing the *Rnf145* gene locus in mouse peritoneal macrophages. ChIP was performed with antibodies against FLAG, or acetylated H3K27 in LXR DKO immortalized mouse bone marrow macrophages reconstituted with FLAG-LXRα or FLAG-LXRβ. *p<0.05; **p<0.01 by student's *t*-test.

DOI: https://doi.org/10.7554/eLife.28766.002

The following figure supplement is available for figure 1:

**Figure supplement 1.** Induction of Rnf145 by LXRs does not require protein synthesis.

DOI: https://doi.org/10.7554/eLife.28766.003

The regulation of *Rnf145* expression by LXR remained intact even with cycloheximide treatment, suggesting that it was a direct transcriptional effect (*Figure 1—figure supplement 1*). This conclusion was corroborated by ChIP-seq analysis showing the binding of both LXRα and LXRβ to the proximal regulatory region of the *Rnf145* gene (*Figure 1E*). ChIP-seq analysis also revealed H3K27 acetylation in response to LXR activation in the same region.

Based on domain structure prediction, RNF145 is postulated to contain 14 transmembrane domains and a RING domain located at the C-terminus. However, the subcellular localization of RNF145 has not been determined. To address this issue, we transfected HEK293 cells with plasmids encoding epitope-tagged RNF145, and performed immunofluorescence staining. We found that the fluorescent signal of the tagged RNF145 protein greatly overlapped with those of the ER-resident proteins calnexin and Sec26, but not with that of Tom20, a mitochondria marker (*Figure 2*). These data strongly suggest that RNF145 is an integral ER membrane protein.

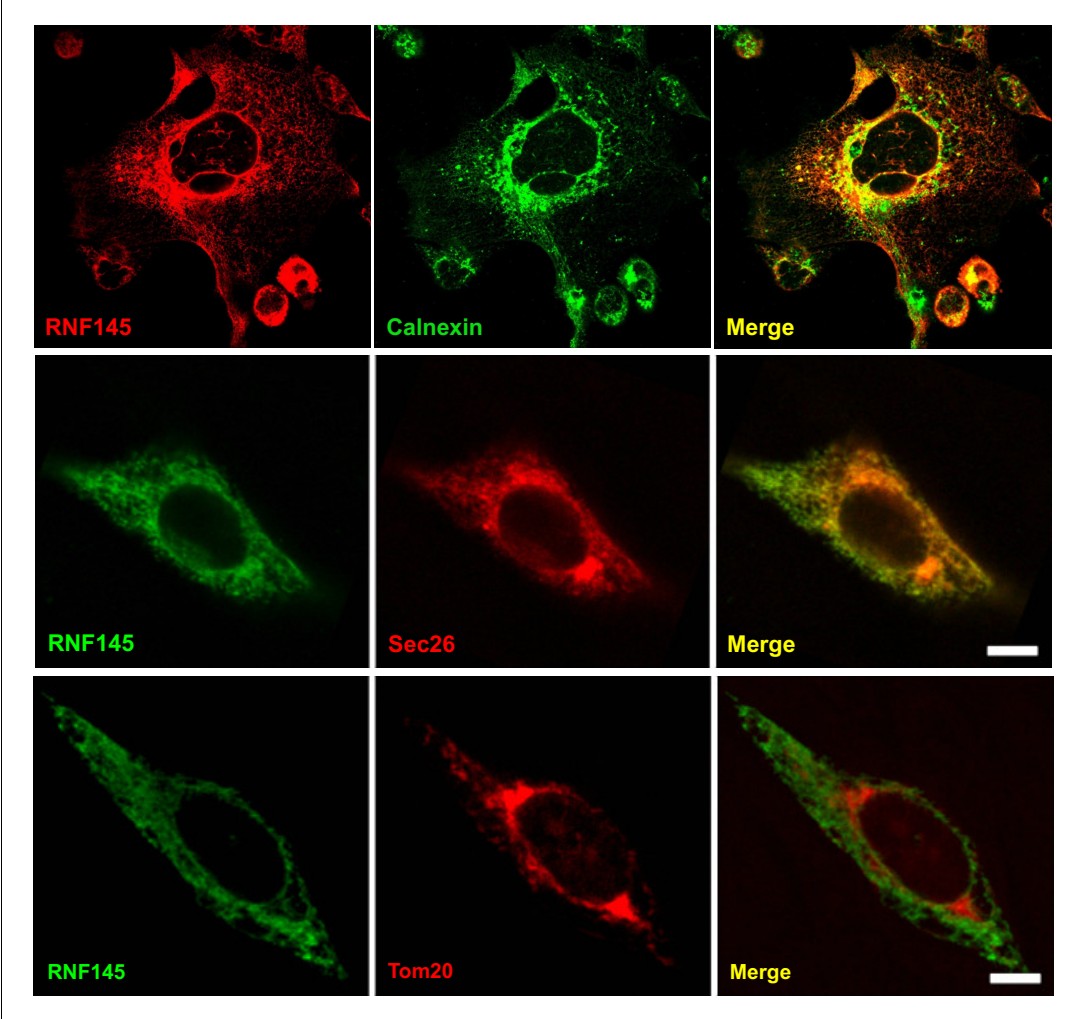

**Figure 2.** Rnf145 encodes an ER-localized ubiquitin E3 ligase. Immunofluorescence staining of HEK293T cells transfected with FLAG-tagged RNF145 with antibodies against the FLAG tag or the ER markers calnexin and Sec26, or the mitochondrial marker Tom20.
DOI: https://doi.org/10.7554/eLife.28766.004

## RNF145 reduces serum cholesterol levels and inhibits cholesterol biosynthesis

To explore the function of RNF145 in vivo, we transduced chow-fed C57BL/6 mice with adenoviruses encoding either *Gfp* control or *Rnf145*. Compared to *Gfp* expression, *Rnf145* expression led to decreased serum cholesterol levels (*Figure 3A*). There was no difference in serum triglyceride levels or in cholesterol content of the liver between groups (*Figure 3—figure supplement 1A and B*). Fractionation of serum lipoproteins demonstrated reduced cholesterol in both the HDL and LDL fractions in mice transduced with *Rnf145* (*Figure 3B*).

In theory either increased hepatic lipoprotein clearance or decreased cholesterol production could lead to the reduction in serum cholesterol levels in response to *Rnf145* expression (*Horton et al., 2002*). To assess the contribution of LDLR-dependent lipoprotein clearance to the actions of RNF145, we transduced chow-fed $Ldlr^{-/-}$ mice with adenoviruses expressing *Gfp* control or *Rnf145*. As expected, basal level of serum cholesterol were markedly elevated in $Ldlr^{-/-}$ mice (*Figure 3C*). Remarkably, *Rnf145* expression retained its ability to lower serum cholesterol levels in $Ldlr^{-/-}$ mice. Indeed, the magnitude of reduction was similar to that observed in C57BL/6 mice. There was no difference in liver cholesterol content between *Gfp*- and *Rnf145*-transduced groups (*Figure 3D*).

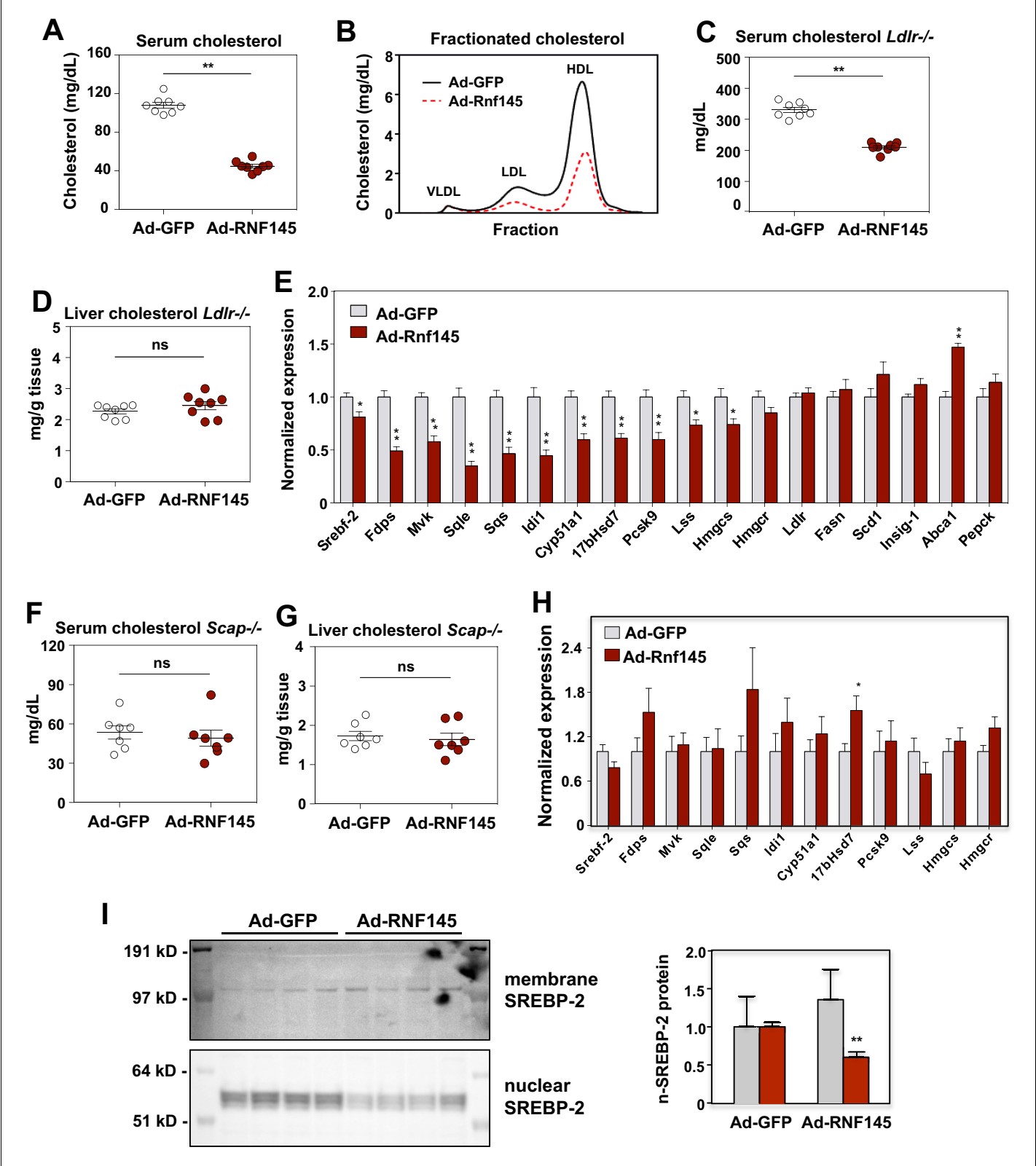

**Figure 3.** RNF145 reduces serum cholesterol levels and inhibits cholesterol biosynthesis. (**A**) Serum cholesterol levels in mice 6 days after injecting with adenovirus expressing GFP or RNF145, N = 8 per group. (**B**) Serum lipoproteins fractionated by FPLC in mice 6 days after inject with adenovirus expressing GFP or RNF145. (**C**) Serum cholesterol levels in *Ldlr*[−/−] mice 6 days after transduction with adenovirus expressing GFP or RNF145. (**D**) Cholesterol content in the liver tissue of *Ldlr*[−/−] mice 6 days after transduction. (**E**). Real-time PCR analysis of gene expression in the liver tissue of mice 6

*Figure 3 continued on next page*

*Figure 3 continued*

days after transduction with adenovirus expressing GFP or RNF145. N = 8 per group. (F) Serum cholesterol levels in liver-specific *Scap* knockout (*Scap*$^{-/-}$) mice 6 days after transduction with adenovirus expressing GFP or RNF145. N = 8 per group. (G) Cholesterol content in the liver tissue of liver-specific *Scap*$^{-/-}$ mice 6 days after transduction. (H) Expression levels of the indicated genes in the liver tissue of liver-specific *Scap*$^{-/-}$ mice 6 days after transduction with adenovirus expressing GFP or RNF145. N = 8 per group. (I) Immunoblot analysis of membrane and nuclear SREBP-2 protein levels in livers from mice transduced with adenovirus expressing GFP or RNF145. Densitometry quantification of the blot is shown at right. *p<0.05; **p<0.01 by student's *t*-test.

DOI: https://doi.org/10.7554/eLife.28766.005

The following figure supplement is available for figure 3:

**Figure supplement 1.** Effects of RNF145 expression in LDLR- or SCAP-deficient mice.
DOI: https://doi.org/10.7554/eLife.28766.006

We next considered the possibility that reduced serum cholesterol levels in the setting of RNF145 expression could reflect suppressed hepatic cholesterol biosynthesis. In line with this idea, we found that the expression of genes involved in cholesterol production was substantially inhibited after 6 days in the liver tissue of mice transduced with *Rnf145* (*Figure 3E*). By contrast, the expression of genes involved in other biological processes, including fatty acid biosynthesis, cholesterol efflux and gluconeogenesis, was not altered on Day 6. The expression of cholesterologenic genes was also suppressed by *Rnf145* in *Ldlr*$^{-/-}$ mice (*Figure 3—figure supplement 1C*), suggesting that LDLR-mediated lipoprotein clearance was not required for these effects of RNF145 on hepatic gene expression.

## Regulation of cholesterol biosynthesis by RNF145 requires intact SREBP signaling

SCAP is essential for generating mature nuclear SREBP transcription factors, and loss of SCAP expression blocks the activity of the SREBP signaling pathways (*Sakai et al., 1998*). To assess whether the activity of hepatic SREBP signaling contributed to the effects of RNF145 on serum cholesterol levels, we transduced chow-fed liver-specific *Scap*-knockout (*Scap*$^{-/-}$) mice with adenoviruses expressing *Gfp* control or *Rnf145*. We found that *Rnf145* expression did not alter serum cholesterol or triglyceride levels in liver-specific *Scap*$^{-/-}$ mice (*Figure 3F* and *Figure 3—figure supplement 1D*). There was also no change in liver cholesterol or triglyceride content with *Rnf145* expression in liver-specific *Scap*$^{-/-}$ mice (*Figure 3G* and *Figure 3—figure supplement 1E*). Furthermore, in liver-specific *Scap*$^{-/-}$ mice, *Rnf145* expression failed to inhibit cholesterologenic gene expression (*Figure 3H*). These data indicate that reduction of serum cholesterol and inhibition of cholesterol biosynthesis by RNF145 requires an intact SREBP signaling pathway. To test whether *Rnf145* expression could alter endogenous SREBP-2 processing in mice, we prepared nuclear fractions from the liver tissue of mice transduced with adenovirus expressing *Rnf145*. We found that *Rnf145* overexpression reduced the levels of the mature, nuclear form of SREBP-2, without affecting the membrane-bound precursor form (*Figure 3I*).

## RNF145, GP78 and TRC8 differentially regulate hepatic metabolism

Several other ER-resident ubiquitin E3 ligases, including GP78 and TRC8, have been demonstrated to participate in the regulation of cholesterol metabolism. The mechanism of action for GP78 and TRC8 has been proposed to involve reductions in the stability of key proteins in the cholesterol biosynthetic pathway, including INSIGs and HMG-CoA reductase (*Irisawa et al., 2009*; *Jo et al., 2011*; *Lee et al., 2010*; *Liu et al., 2012*; *Song et al., 2005*). To begin to address whether RNF145 shared a common mechanism with either of these E3 ligases, we compared the effects of RNF145, GP78 and TRC8 in mouse liver. We transduced chow-fed C57BL/6 mice with adenovirus expressing GFP control, RNF145, GP78 or TRC8. Interestingly, only the expression of RNF145 led to reduced cholesterol levels in the serum (*Figure 4A*). There was no change in serum or liver triglyceride levels in mice transduced with any of the ligases. Although the expression of RNF145 did not alter cholesterol content in the liver, expression of GP78 or TRC8 led to an increase in liver cholesterol content (*Figure 4A*). In line with these effects on lipid levels, whereas RNF145 inhibited the expression of cholesterologenic genes, GP78 and TRC8 elevated expression of the genes involved in both cholesterol and triglyceride production (*Figure 4B*). These effects of GP78 and TRC8 are consistent with

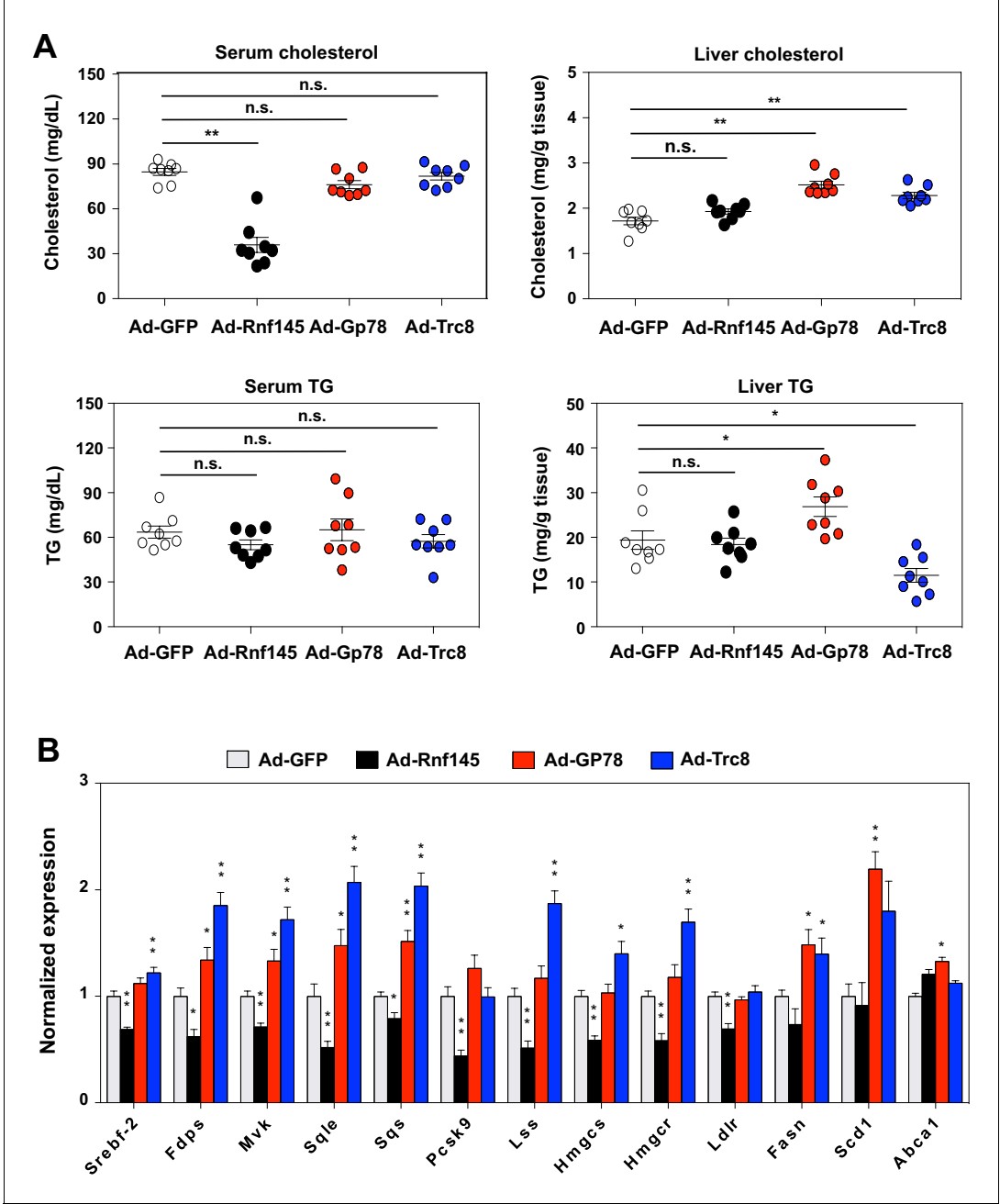

**Figure 4.** Differential effects of hepatic E3 ubiquitin ligases on metabolism. (**A**) Serum and liver cholesterol and triglyceride levels in mice 6 days after transduction with adenovirus expressing GFP, RNF145, GP78 or TRC8. N = 8 per group. (**B**) Real-time PCR analysis of gene expression in liver tissue from mice 6 days after transduction with adenovirus expressing GFP, RNF145, GP78 or TRC8. N = 8 per group. *p<0.05; **p<0.01.
DOI: https://doi.org/10.7554/eLife.28766.007

previous reports that they target INSIGs and HMG-CoA reductase for degradation. The differential effects of RNF145 compared to the other ER-resident E3 ubiquitin ligases strongly suggested that it was affecting sterol metabolism through a distinct mechanism, likely by targeting distinct protein for ubiquitination.

## RNF145 ubiquitinates SCAP to inhibit SREBP processing

RNF145 is predicted to encode an E3 ubiquitin ligase with a RING-type catalytic domain. To examine whether the E3 ligase activity of RNF145 was required for its ability to reduce serum cholesterol

levels and inhibit cholesterologenic gene expression, we transduced chow-fed C57BL/6 mice with an adenovirus encoding a mutant version of RNF145 harboring a cysteine mutation within its RING domain (C537A) (*Figure 5—figure supplement 1*). This mutation disrupts the structural stability of the RING domain and therefore renders the E3 ligase catalytically inactive (*Hershko and Ciechanover, 1998*). Interestingly, the expression of mutant *Rnf145* in liver did not alter cholesterol levels in the serum (*Figure 5A*). Furthermore, the hepatic expression of genes involved in cholesterol biosynthesis was not suppressed by the expression of RING-mutant *Rnf145* (*Figure 5B*).

Since the RNF145 effects on serum cholesterol levels and cholesterol biosynthesis were abolished in mice lacking hepatic *Scap* expression, we hypothesized that SCAP, also an ER resident protein, could be the ubiquitination target of RNF145. To test this hypothesis, we transfected CHO cells with plasmids encoding epitope-tagged *Scap* and ubiquitin, together with WT *Rnf145* or RING-mutant *Rnf145* (C537A). We found that WT RNF145, but not the C537A mutant, induced the ubiquitination of SCAP as evidenced by Western blotting with an antibody against ubiquitin (*Figure 6A*). We were unable to detect an interaction of SCAP and RNF145 in co-immunoprecipitation assays (*Figure 6—figure supplement 1A*), although this finding was not unexpected given the transient interactions between many E3 ubiquitin ligases and their substrates (*Harper and Tan, 2012*). Surprisingly, despite the ubiquitination, the abundance of SCAP protein did not change in response to RNF145 expression. We further confirmed using cycloheximide to block protein synthesis that the turnover of SCAP protein was not affected by exogenous expression of RNF145 (*Figure 6—figure supplement 1B*). These data collectively suggest that the ubiquitination induced by RNF145 does not direct SCAP to degradation.

Since RNF145 inhibited cholesterologenic gene expression known to be controlled by SREBP-2 and reduced nuclear abundance of mature SREBP-2 in mouse liver, we next sought to determine whether RNF145 affected the processing of SREBP-2. We transfected HEK293 cells with plasmids encoding HSV-tagged SREBP-2 under the control of a TK promoter, together with GFP control, WT RNF145, mutant RNF145 (C537A), GP78, or TRC8. Sterol depletion of the cell culture medium induced efficient SREBP-2 processing, as evidenced by increased accumulation of the cleaved nuclear form of SREBP-2 (*Figure 6B*, *Figure 6—figure supplement 1C*). In cells transfected with WT RNF145, sterol depletion-induced processing of SREBP-2 was substantially inhibited. In

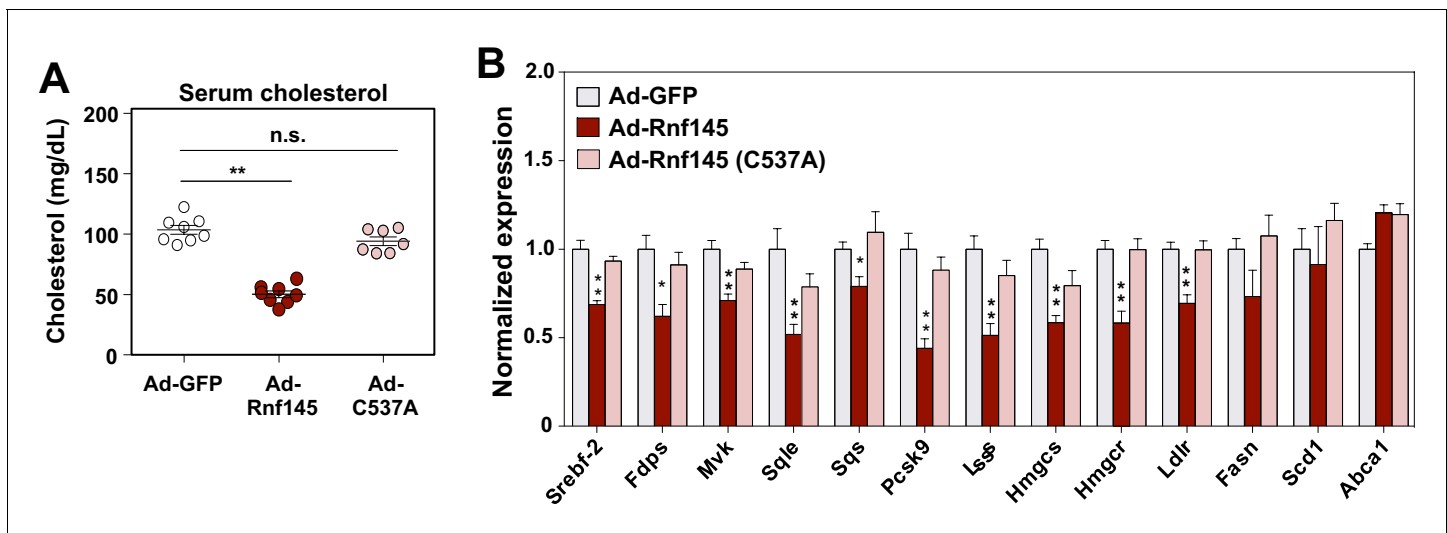

**Figure 5.** The E3 ligase activity of RNF145 is required for its effects on hepatic metabolism. (**A**) Serum cholesterol levels in mice 6 days after transduction with adenovirus expressing GFP, RNF145, or RING-mutant RNF145 (C537A). N = 8 per group. (**B**) Real-time PCR analysis of gene expression in the liver tissue of mice 6 days after adenoviral transduction. N = 8 per group. *p<0.05; **p<0.01.

DOI: https://doi.org/10.7554/eLife.28766.008

The following figure supplement is available for figure 5:

**Figure supplement 1.** Real-time PCR analysis of gene expression in the liver tissue of mice 6 days after transduction with adenovirus expressing GFP, RNF145, or RING-mutant RNF145 (C537A).
DOI: https://doi.org/10.7554/eLife.28766.009

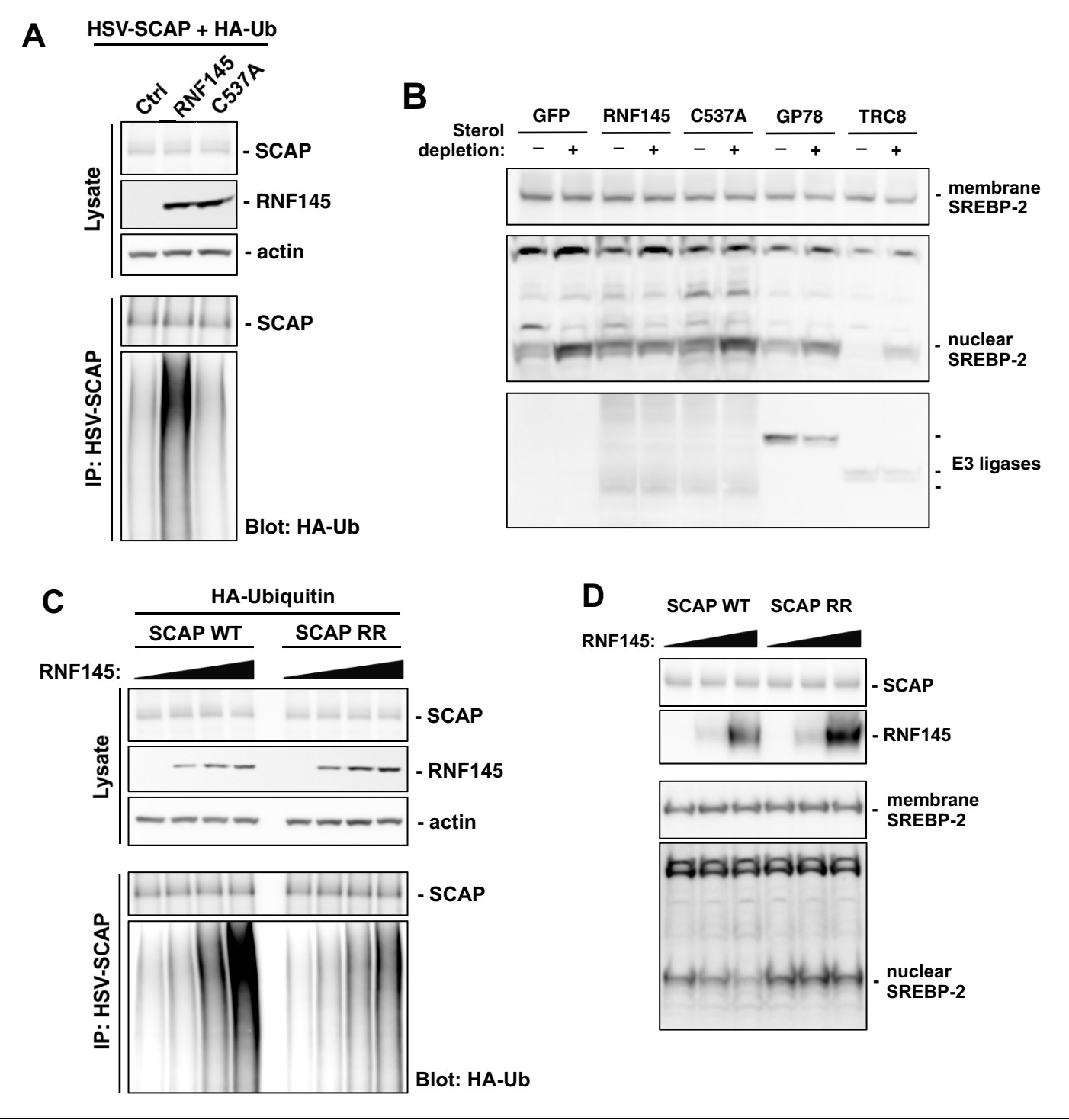

**Figure 6.** RNF145 ubiquitinates SCAP to inhibit SREBP processing. (**A**) CHO cells were transfected with plasmids encoding HSV-tagged SCAP and HA-tagged ubiquitin, together with RNF145 or RING-mutant RNF145 (C537A). Cell lysates were blotted with the indicated antibodies. Cell lysates were also immunoprecipitated with HSV antibody, and the precipitated proteins were blotted with HA antibody. (**B**) HEK293T cells were transfected with plasmids encoding HSV-tagged SREBP-2 under the control of a TK promoter, together with GFP control, RNF145 or RING-mutant RNF145, GP78, or TRC8. Cells were cultured in DMEM medium supplemented with 10% FBS, or in DMEM medium supplemented with 10% lipoprotein-deficient serum and 25 μM simvastation overnight to deplete sterol, and then treated for 4 hr with MG132 (25 μM). Membrane and nuclear protein fractions of the cells were then isolated and blotted with the indicated antibodies. (**C**) CHO cells were transfected with plasmids encoding RNF145 and HA-tagged ubiquitin, together with HSV-tagged SCAP or mutant SCAP (K454R, K466R). Cell lysates were blotted with the indicated antibodies. Cell lysates were also

*Figure 6 continued on next page*

Figure 6 continued

immunoprecipitated with HSV antibody, and the precipitated proteins were blotted with HA antibody. (**D**) *SCAP*-knockout HEK293 cells were transfected with plasmids encoding HSV-tagged SREBP-2 under the control of a TK promoter and RNF145, together with V5-tagged SCAP or ubiquitination-site mutant SCAP (K454R, K466R). Cells were cultured in DMEM medium supplemented with 10% lipoprotein-deficient serum and 25 μM simvastatin overnight to deplete sterol, and then treated for 4 hr with MG132 (25 μM). Membrane and nuclear protein fractions were isolated and blotted with the indicated antibodies.

DOI: https://doi.org/10.7554/eLife.28766.010

The following figure supplement is available for figure 6:

**Figure supplement 1.** Interaction between RNF145 and the SCAP pathway.

DOI: https://doi.org/10.7554/eLife.28766.011

contrast, the sterol-regulated processing of SREBP-2 in cells receiving mutant RNF145, GP78, or TRC8, was comparable to controls (*Figure 6B*).

Prior work has defined amino acid residues within cytoplasmic loop 6 of SCAP that are essential for its COPII binding, and thus required for the ER to Golgi transport and subsequent processing of SREBP-2 (*Sun et al., 2005*). We noticed that there were two lysine residues (K454 and K466) positioned in proximity to the residues important for COPII binding within the same cytoplasmic loop (*Figure 6—figure supplement 1D*). We postulated that ubiquitination on these lysine residues induced by RNF145 might affect the transport and processing of SREBP-2. To test this hypothesis, we mutated these SCAP lysine residues to arginine (K454R, K466R) so that they could not be ubiquitinated. When we transfected the mutant SCAP into CHO cells together with RNF145, we found that the ubiquitination of the mutant SCAP was reduced compared to WT SCAP (*Figure 6C*), suggesting that RNF145 was targeting these lysine residues.

To further investigate the involvement of SCAP ubiquitination in the effects of RNF145 on SREBP processing, we reconstituted *SCAP*-knockout HEK293 cells (*Shao et al., 2016*) with either WT SCAP or lysine-mutant SCAP (K454R, K466R). Under sterol depletion conditions, we found that *Rnf145* expression was able to inhibit SREBP-2 processing in *SCAP*-knockout cells reconstituted with WT SCAP, but not in those reconstituted with lysine-mutant SCAP (*Figure 6D*). These results suggest that ubiquitination of SCAP on lysine residues K454 and K466 is the basis for RNF145-mediated inhibition of SREBP-2 processing.

Since SCAP is a key regulator of the processing of both SREBP-1 and SREBP-2, our finding of preferential effects of RNF145 on SREBP-2 pathway genes 6 days after transduction was unexpected. A more detailed time course of SREBP target gene expression from 2 to 6 days post-RNF145 transduction revealed that both SREBP-1 and SREBP-2 targets were initially suppressed by RNF145, but that SREBP-1 targets recovered over time (*Figure 6—figure supplement 1E*). The basis for this differential response of SREBP-1 and SREBP-2 target genes is currently unknown.

## Acute or chronic RNF145 deficiency affects hepatic sterol synthesis

To explore the physiological role of endogenous RNF145 in the maintenance of cholesterol homeostasis, we first transduced chow-fed C57BL/6 mice with an adenovirus encoding an shRNA targeting *Rnf145*. Endogenous hepatic *Rnf145* mRNA expression levels were suppressed by about 40% 6 days post transduction with shRNA (*Figure 7A*). We did not observe differences in serum cholesterol or triglyceride levels with this partial *Rnf145* knockdown (*Figure 7B*). However, the cholesterol content in the liver of mice expressing *Rnf145*-targeting shRNA was modestly increased. Furthermore, the expression of a number of cholesterologenic genes was elevated in response to reduced *Rnf145* expression, while genes involved in other biological processes, including cholesterol efflux were largely unchanged (*Figure 7B*).

We also generated *Rnf145*-knockout mice (*Rnf145⁻/⁻*) by CRISPR/Cas9-mediated gene editing to assess the effects of chronic inactivation of *Rnf145* expression (*Figure 8A*). These mice were born at Mendelian ratios and were externally unremarkable except for a slightly reduced body weight compared to WT littermates (*Figure 4B*). Interestingly, serum cholesterol levels in *Rnf145⁻/⁻* mice were increased in both the LDL and HDL fractions compared to their WT littermates (*Figure 4B and C*). Liver cholesterol and triglyceride levels, and blood glucose levels were not altered in *Rnf145⁻/⁻* mice (*Figure 4B*).

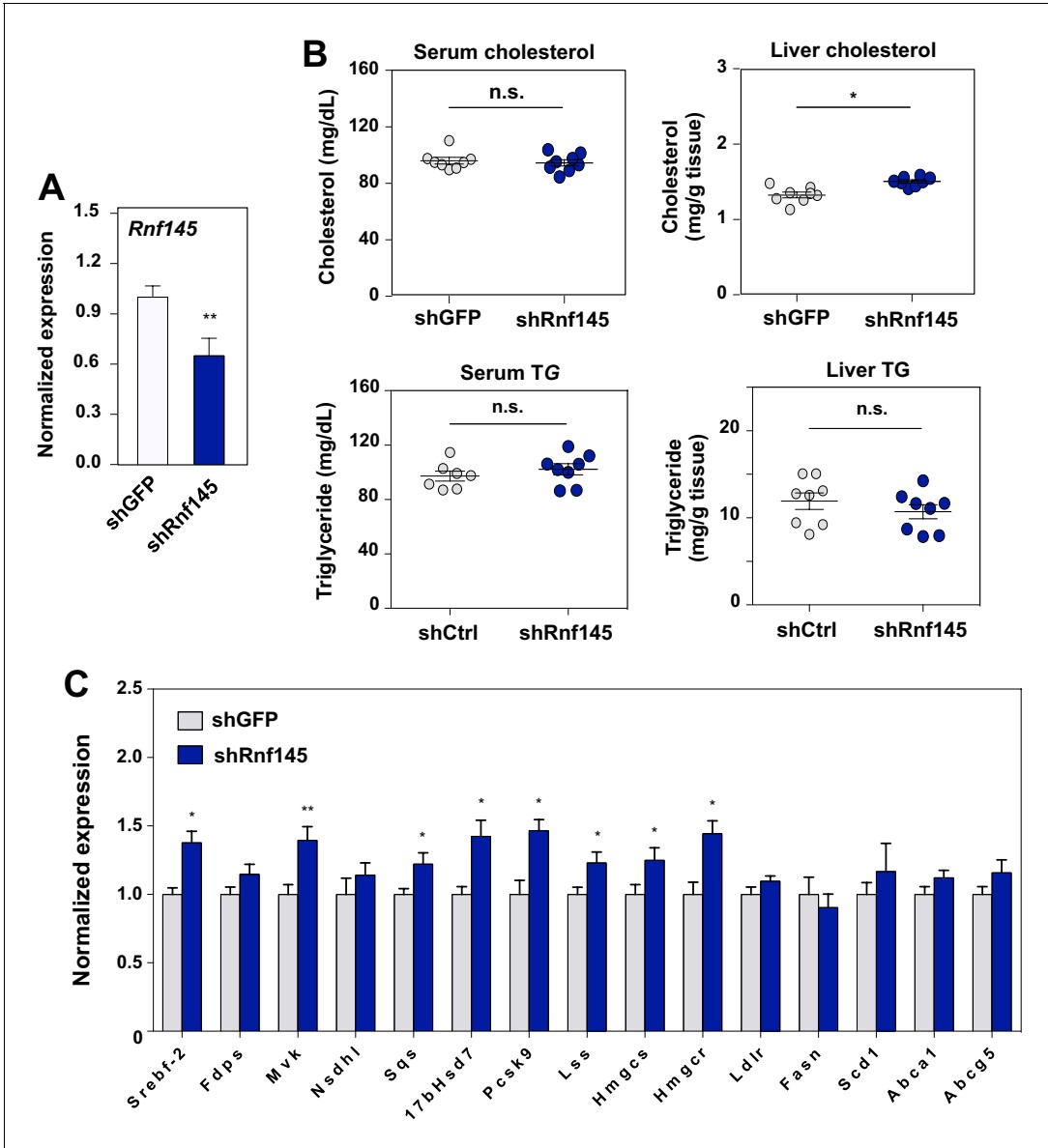

**Figure 7.** Acute deficiency of RN 145 modulates hepatic sterol metabolism. (**A**) Expression levels of *Rnf145* in mice 6 days after injecting with adenovirus expressing control shRNA or shRNA targeting *Rnf145*. N = 8 per group. (**B**) Serum cholesterol levels in mice in A. (**C**) Cholesterol content in the liver tissue of mice in A. (**D**) Expression levels of the indicated genes in the liver tissue of mice 6 days after transduction with adenovirus expressing control shRNA or shRNA targeting *Rnf145*. N = 8 per group.

DOI: https://doi.org/10.7554/eLife.28766.012

To assess the global impact of RNF145 deficiency for hepatic gene expression, we performed transcriptional profiling of WT and *Rnf145*[−/−] livers. Unbiased gene ontology and pathway analysis revealed that the cholesterol biosynthetic pathway was coordinately upregulated in *Rnf145*[−/−] mice compared to controls (*Figure 4D*). These results were further validated by real-time PCR (*Figure 4E*). Expression of a number of SREBP-2 target genes involved in the sterol synthesis pathway was increased in the genetic absence of *Rnf145*. These changes were reciprocal to those observed with RNF145 overexpression (cf. *Figure 3E*). In sum, both acute suppression of RNF145 via shRNA-mediated knockdown and chronic inactivation of RNF145 by genetic deletion led to increased hepatic sterol production. These data suggest endogenous *Rnf145* participates in the maintenance of cholesterol homeostasis by modulating cholesterol biosynthesis.

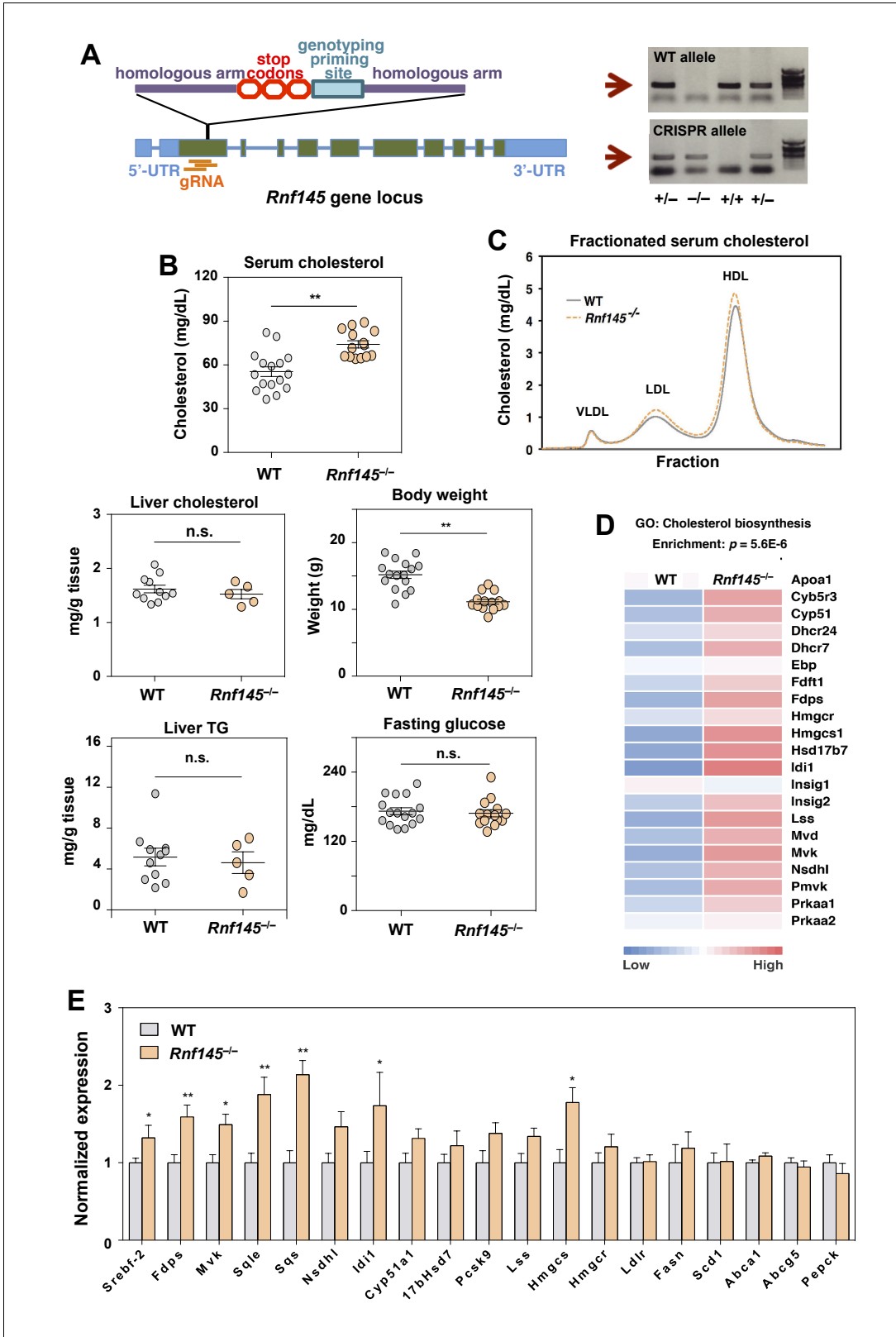

**Figure 8.** Genetic deficiency of Rnf145 affects cholesterol metabolism. (**A**) Schematic diagram showing the strategy used to generate the *Rnf145* knockout mouse line (*Rnf145*[−/−]) by CRISPR/Cas9-mediated gene editing. (**B**) Serum and liver cholesterol levels, liver triglyceride levels, body weight and fasting blood glucose in 4 week-old *Rnf145* knockout (*Rnf145*[−/−]) mice and WT littermates following a 6 hr fast. N = 16 and 14 for WT and *Rnf145*[−/−] mice, respectively. (**C**) Cholesterol levels in FLPC-fractionated serum from mice in B. (**D**) Heatmap representation of the expression levels of genes

*Figure 8 continued on next page*

*Figure 8 continued*

categorized to the cholesterol biosynthetic pathway from microarray-based gene expression profiling of the liver tissue of 4 week-old *Rnf145⁻/⁻* mice and WT littermates following a 6 hr fast. (E) Real-time PCR analysis of gene expression in the liver tissue of 4 week-old *Rnf145⁻/⁻* mice and WT littermates following a 6 hr fast. N = 11 and 5 for WT and *Rnf145⁻/⁻* mice, respectively. *p<0.05; **p<0.01 by student's *t*-test.

DOI: https://doi.org/10.7554/eLife.28766.013

## Discussion

In cells and organisms, cholesterol levels are kept within a very narrow range. In order to achieve this goal, two signaling axes with opposing functions – the SREBP and LXR signaling pathways – must work in a coordinated fashion (*Hong and Tontonoz, 2014*; *Horton et al., 2002*). Despite our understanding of the mechanics underlying these individual pathways, modes of crosstalk between them have only recently been explored (*Rayner et al., 2010*; *Sallam et al., 2016*; *Wong et al., 2006*; *Zelcer et al., 2009*). In this study, we identified RNF145 as LXR- and sterol-induced factor that plays a role in the response to elevated hepatic cholesterol load. Gain or loss of RNF145 function in mice affects serum cholesterol levels and the expression of hepatic genes involved in sterol synthesis. The demonstration that RNF145 is a functional E3 ligase that modulates SREBP-2 processing through ubiquitination of SCAP outlines an additional mechanism for coordination of LXR and SREBP actions in lipid homeostasis.

Over the past decade, ubiquitination has emerged as an important mode of post-transcriptional regulation of cellular lipid homeostasis. We previously identified Idol as an E3 ligase that targets LDLR for lysosomal degradation (*Zelcer et al., 2009*). Several ubiquitin E3 ligases localized in the ER have also been demonstrated to participate in the regulation of cholesterol biosynthesis. GP78 and TRC8 have been shown to mediate sterol-induced degradation of a rate-limiting enzyme in cholesterol biosynthesis, HMG-CoA reductase (*Jo et al., 2011*; *Song et al., 2005*). In addition, these two E3 ligases have been reported to affect the activity of the SREBP-2 pathway (*Irisawa et al., 2009*; *Lee et al., 2010*; *Liu et al., 2012*). GP78 degrades INSIG to stimulate SREBP activity (*Liu et al., 2012*; *Song et al., 2005*). TRC8 was reported to reduce the stability of SREBP, and to inhibit SREBP processing in cultured cells (*Irisawa et al., 2009*; *Lee et al., 2010*). The ER-resident ubiquitin E3 ligase MARCH6 was shown to be involved in the degradation of squalene epoxidase (*Zelcer et al., 2014*). Another group has recently confirmed the regulation of *Rnf145* by LXRs, but did not observe metabolic consequences of acute knockdown of this E3 ligase in vitro (*Cook et al., 2017*). Our analysis of the *Rnf145⁻/⁻* mice has thus revealed phenotypes associated with RNF145 deficiency not predicted by in vitro models.

Although RNF145 shares a certain degree of structural similarity to other ER membrane E3 ligases, especially TRC8, it appears to serve a very distinct function in cholesterol homeostasis. In our study, RNF145 was the only protein among the E3 ligases tested to inhibit the expression of genes involved in cholesterol biosynthesis and to reduce serum cholesterol levels when expressed in mouse liver. Interestingly, analysis of domain structure of RNF145 suggests the protein, like TRC8, may harbor a sterol-sensing domain (*Lee et al., 2010*). This raises the possibility that, in addition to its transcriptional regulation by LXR, the protein abundance or E3 catalytic activity of RNF145 could be subject to post-transcriptional regulation in response to cellular sterol levels.

*Rnf145* was presumed to encode a RING-type ubiquitin E3 ligase based on its sequence, but its potential targets and biological function were previously unknown. In this study we identified SCAP, an important regulator of SREBP signaling, as a direct ubiquitination target of RNF145. This finding is intriguing as there have been few prior reports of post-translational modification of SCAP activity. Bornfeldt and colleagues reported that SCAP was phosphorylated by the cAMP pathway during steroidogenesis (*Shimizu-Albergine et al., 2016*). We have extended this work to show that SCAP can also be ubiquitinated. In most cases, polyubiquitination of proteins with K48-linked chains often serves to target proteins for degradation by the proteasome (*Chau et al., 1989*), whereas K63 linkages have been associated with non-proteolytic functions. Interestingly, ubiquitination of SCAP by RNF145 did not lead to degradation of SCAP in our assays. Non-proteolytic ubiquitination via K63 linkage is particularly common in membrane proteins, whose sorting and transport are frequently regulated by this mode of modification (*Mukhopadhyay and Riezman, 2007*).

In this study, both acute suppression of RNF145 via shRNA-mediated knockdown and chronic inactivation of RNF145 by genetic deletion potentiated the expression of cholesterol biosynthetic genes in the liver. However, the effects of these different manipulations of RNF145 expression were only partially overlapping. For example, acute knockdown of *Rnf145* led to an elevation of liver cholesterol content, but this effect was not observed in the chronic knockout model. One plausible explanation for this difference is that chronic activation of SREBP signaling could lead to the activation of compensatory machinery that ameliorates sterol accumulation in the liver.

Since SCAP is a key regulator of the processing of both SREBP-1 and SREBP-2, alterations of SCAP might be expected to affect both SREBP-1 and -2 target genes. We observed suppression of both SREBP-1 and -2 target genes in response to acute RNF145 expression in the liver, but SREBP-1 targets recovered of the course of several days whereas the SREBP-2 pathway remained suppressed. In the setting of developmental deletion loss of RNF145 (*Rnf145*$^{-/-}$ mice), we observed selective upregulation of SREBP-2 target genes. The basis for this differential response is not yet clear, but compensatory regulatory mechanisms appear likely to be at play. It is known that the interactions of the INSIG-SCAP complex with SREBP-1 and SREBP-2 are differentially sensitive to sterols, as increasing hepatic sterol content leads to decreased SREBP-2 but increased SREBP-1 pathway activity (*Repa et al., 2000a*). It is possible that ubiquitination of SCAP differentially affects its interactions with SREBP-1 and SREBP-2.

How does the LXR-RN145-SCAP pathway fit with the INSIG-SCAP sterol-sensing pathway for SREBP-2 regulation? It is clear from studies of knockout mice that the INSIG-SCAP pathway is the dominant regulator, especially in the liver. However, redundant mechanisms have evolved for the modulation of many important biological pathways. In fact several examples of parallel or complementary mechanisms for regulation of metabolism by SREBPs and LXR have been described. LXRs are critical for maximal transcription of SREBP-1c, and loss of hepatic LXR expression therefore compromises SREBP-1c-dependent lipogenesis (*DeBose-Boyd et al., 2001*; *Repa et al., 2000a*; *Schultz et al., 2000*). LXR activation also promotes the proteolytic processing of SREBP-1c, but not SREBP-2, by altering membrane phospholipid composition though induction of *Lpcat3* (*Rong et al., 2017b*). Oxysterols not only inhibit SREBP processing, but also activate LXRs. Indeed, recent work from Horton and colleagues indicates that sterol LXR ligands generated by the SREBP-2 pathway are actually required for SREBP-1c pathway activity in the liver (*Rong et al., 2017a*). The relative abilities of specific sterols to regulate SREBP and LXR also differ. For example, 25-hydroxycholesterol is a potent inhibitor of SREBP processing but a weak LXR agonist, whereas 22(R)-hydroxycholesterol preferentially activates LXR (*Scotti et al., 2011*). Therefore, RNF145 action is likely to be more relevant when LXR activators are abundant. Unfortunately, as the recent Horton study also makes clear, the identities of the key endogenous LXR activators in liver remain to be fully determined (*Rong et al., 2017a*).

Long-term investigations of transgenic and knockout models, performed under a variety of dietary conditions, will be needed in order to fully define the physiological role of the LXR-RNF145 pathway in metabolism. Given, that RNF145 expression is abundant in other metabolic tissues such as the intestine, it seems likely that tissue-specific loss-of-function models will also be informative. Nevertheless, the present study has delineated a previously-unrecognized mechanism for ubiquitin-mediated crosstalk between the LXR and SREBP signaling pathways and opened up new avenues for future study. It is tempting to speculate that manipulation of RNF145 activity could provide an alternative strategy for pharmacological intervention in human cholesterol metabolism. This could be especially attractive in settings in which the LDLR receptor is inactive and statins are ineffective, such as familial hypercholesterolemia.

## Materials and methods

### Reagents, plasmids, and gene expression

GW3965 was synthesized as previously described (*Collins et al., 2002*). LG268 was from Ligand Pharmaceuticals. Ligands were dissolved in dimethyl sulfoxide before use in cell culture. Simvastatin sodium salt was from Calbiochem. MG132 was from Sigma.

For transient transfections and viral vector production of the ubiquitin E3 ligases, the sequences encoding RNF145, GP78 and TRC8 ere amplified from cDNA reversely transcribed from mouse total

mRNA. The resulting sequence fragments were tagged with a FLAG polypeptide sequence (DYKDDDDK) at the C-terminus, and then cloned in to pDONR221 entry vector using the Gateway system. The C537A mutation of *Rnf145* was introduced using the Stratagene Quickchange kit. The sequences in the entry clones were then transferred by LR recombination into pcDNA-DEST47 for transient transfections, or into pAd/CMV/V5-DEST Gateway vector for viral particle production. pTK-HSV-SREBP-2 was a gift from Drs. Michael Brown and Joseph Goldstein (University of Texas Southwestern Medical Center, Dallas, Texas, USA). To construct the pTK-V5-hmSCAP plasmid for transient transfections, the hamster SCAP sequence was amplified from the pTK-HSV-hmSCAP plasmid (a gift from Drs. Michael Brown and Joseph Goldstein). The resulting sequence was tagged with a V5 polypeptide sequence (GKPIPNPLLGLDST) at the N terminus, and then cloned into pre-digested pTK-HSV-SREBP-2 plasmid between the *Spe*I and *Pac*I sites. The K454R and K466R mutations were introduced using the Stratagene Quickchange kit. To obtain the sh*Rnf145* adenovirus, Invitrogen based software was used to generate nucleotide sequences targeting the mouse *Rnf145* gene. The nucleotide sequences were then cloned into pENTR/U6. The resulting pENTR/U6-*Rnf145* shRNA plasmids were tested for their ability to inhibit overexpressed *Rnf145* in transient transfection experiments in HEK293T cells and then transferred by LR recombination into the pAd/BLOCK-iT-DEST destination vector for viral particle production. The following sh*Rnf145* nucleotide sequences were used: 5'-GCCTCAACAACAACCCTCTTT -3'. All adenoviruses were amplified, purified, and titered by Viraquest.

For gene expression analysis, RNA was isolated using TRIzol reagent (Invitrogen) and reversely transcribed into cDNA. Real-time PCR was performed using an Applied Biosystems 7900HT sequence detector or Applied Biosystems Quant Studio 6 Flex. Results were normalized to the average expression levels of 36B4, actin and cyclophilin.

## Animals and diets

All animals were housed in a temperature-controlled room under a 12 hr light–dark cycle and under pathogen-free conditions. Mice were placed on a chow diet except as indicated where mice were fed a Western diet (21% fat, 0.21% cholesterol; D12079B; Research Diets Inc.). LXRα-deficient, LXRβ-deficient, and LXR DKO mice were originally provided by David Mangelsdorf (University of Texas Southwestern Medical Center, Dallas, Texas, USA). Floxed $Scap^{-/-}$ mice were crossed with Albumin-*Cre* mice at UCLA (*Tarling et al., 2015*). For adenovirus experiments age-matched wild type and $Ldlr^{-/-}$ mice were purchased from Jackson Laboratories. Littermates in all experiments were manually randomized to different treatment groups. Investigators were blinded to group allocation for some but not all studies. *Rnf145* global knockout mice were generated at UC Irvine facility using the CRISPR/Cas9 strategy outlined in *Figure 8A*. For adenoviral infections, age-matched (8–10 weeks old) male mice were injected with $2.0 \times 10^9$ PFU by tail-vein injection. Serum cholesterol and triglyceride levels were measured as previously described (*Hong et al., 2012*). Mice were sacrificed 2 to 6 days later following a 6 hr fasting period. At the time of sacrifice liver tissue and blood collected by cardiac puncture were immediately frozen in liquid nitrogen and stored at −80°C. Liver tissue was processed for isolation of RNA and protein as described previously (*Hong et al., 2014*). All animal experiments were approved by the UCLA Institutional Animal Care and Research Advisory Committee.

## Cell culture and transfection

HEK293T and CHO cells were obtained from the American Type Culture Collection. They have been previously verified by STR testing and were confirmed to be mycoplasma-free by regular testing. Cells were cultured in Dulbecco's Modified Eagle medium supplemented with 10% fetal bovine serum (Invitrogen), 100 U/ml penicillin, and 100 U/ml streptomycin. Cells were incubated at 37°C in a humidified incubator containing 5% $CO_2$. *SCAP*-knockout HEK293T cells were kindly provided by Dr. Peter Espenshade (Johns Hopkins University School of Medicine, Baltimore, MD, USA), and were cultured in the above-mentioned culture medium additionally supplemented with 5 µg/ml cholesterol, 1 mM sodium mevalonate, and 20 mM oleic acid. All cells were transfected with Fugene 6 (Promega) following manufacturer's protocol.

## Protein isolation and western blot

Nuclei from mouse liver tissues were prepared as previously described (*Jeon et al., 2013*; *Tarling et al., 2015*). Total cellular and tissue proteins were extracted with RIPA lysis buffer (Thermo Scientific) with proteinase/phosphatase inhibitors (Thermo Scientific). Lysate was separated by 4–12% sodium dodecyl sulfate polyacrylamide gel electrophoresis (SDS-PAGE). Proteins on the gel were electrophoretically transferred onto nitrocellulose membranes (Amersham Biosciences). The membranes were blocked in PBS supplemented with 5% non-fat milk to quench nonspecific protein binding, and then incubated with the indicated antibodies. Horse radish peroxidase-conjugated anti-mouse and anti-rabbit IgG (Jackson) were used as secondary antibodies. The immune signal was visualized using the ECL kit (Amersham Biosciences).

The following primary antibodies were used: anti-FLAG tag M2 antibody (Sigma, F3165), anti-HA tag antibody (Covance, MMS-101R), anti-V5 tag antibody (Life Technologies, R960-25), anti-actin antibody (Sigma, A2066), anti-HSV tag antibody (Millipore, 69171), anti-calnexin antibody (Abcam, ab10286), anti-SREBP-2 antibody (Millipore, MABS1988).

## Immunofluorescence staining

Cells cultured on coverslips were washed three times with cold PBS and fixed in 100% methanol at −20°C for 5 min. Cells were then permeabilized in PBS with 0.3% Triton X-100% and 0.15% BSA for 15 min at room temperature and blocked in goat serum dilution buffer (16% goat serum, 20 mM sodium phosphate, pH 7.4, 450 mM NaCl, 0.3% Triton X-100) for 30 min at room temperature. Cells were incubated in primary antibody diluted 1:200 in blocking buffer for 1 hr at room temperature, then immersed three times in permeabilization buffer, and then incubated for 1 hr in secondary antibody diluted 1:200 in blocking buffer. Cells were then rinsed three times in PBS before mounting with Vectashield (Vector Laboratories). Cells were visualized on a Zeiss LSM 510 META confocal microscope.

## Immunoprecipitation

Cells were lysed in RIPA buffer supplemented with protease inhibitors. The concentration of proteins was determined with the BCA assay method. Equal amounts of protein were incubated with Protein A/G sepharose beads (Santa Cruz Biotech) and the indicated primary antibodies at 4°C overnight. The mixture was then centrifuged at 2000 × g at 4°C for 2 min to pellet the beads. The pellet was then washed with RIPA buffer for three times. Immunoprecipitated proteins on the beads were washed off using SDA-PAGE loading buffer, with incubation at 55°C for 15 min.

For ChIP studies, immortalized bone marrow-derived macrophages (iBMDM) from LXR DKO mice (*Ito et al., 2015*) were reconstituted with N-terminus 3xFLAG-tagged LXRα or LXRβ using the pBabe retroviral expression system. A control line was also prepared by transduction with an empty pBabe-puro vector. Macrophages (12 × 10^6) were crosslinked using a double fixation protocol with 2 mM disuccinimidyl glutarate for 30 min and 1% methanol-free ultrapure formaldehyde for 10 min before quenching with 2 M glycine. Cells were lysed with RIPA buffer and, after chromatin shearing by sonication, incubated overnight with protein G magnetic Dynabeads (Invitrogen) coupled with 3 µg of either anti-FLAG M2 (SIGMA #F3165) or anti-H3K27ac (Abcam #ab4729) antibodies. For high-throughput sequencing, a minimum of 10 ng of DNA was obtained by pooling DNA from 10 independent ChIP reactions (for FLAG-LXR sequencing) or six different ChIP reactions (for H3K27ac sequencing). DNA was then used for library preparation and subsequent Illumina HiSeq sequencing by the Centre de Regulació Genomica (CRG, Barcelona, Spain) genomic facility.

## Nuclear and membrane preparation

Cultured cells were collected in Buffer B (10 mM Hepes-KOH at pH7.4, 10 mM KCl, 1.5 mM MgCl$_2$, 0.5 mM EDTA sodium, 0.5 mM EGTA sodium, 1 mM DTT, protease inhibitors) and incubated on ice for 10 min. Cell suspension was then passed through a 23-gauge needle for 15 times to break the cellular membrane structures. Cell suspension was then centrifuged at 1000 × g at 4°C for 5 min. The resulting 1000 × g pellet was the nuclear fraction. The nuclear fraction was resuspended and incubated on ice for 30 min with intermittent agitation in Buffer C (10 mM Hepes-KOH at pH7.4, 0.42 M NaCl, 2.5% (v/v) glycerol, 1.5 mM MgCl$_2$, 0.5 mM EDTA sodium, 0.5 mM EGTA sodium, 1 mM DTT, protease inhibitors), followed by 10,000 × g centrifugation at 4°C for 10 min. The

10,000 × g supernatant was the nuclear protein extraction. The 1000 × g supernatant was used for membrane fraction preparation. The 1000 × g supernatant was centrifuged at 100,000 × g at 4°C for 30 min. The resulting pellet was washed briefly with Buffer A (10 mM Hepes-KOH at pH7.4, 10 mM KCl, 1.5 mM $MgCl_2$, 0.5 mM EDTA sodium, 0.5 mM EGTA sodium, 100 mM NaCl) and then lysed in RIPA buffer supplemented with protease inhibitors, followed by 10,000 × g centrifugation at 4°C for 10 min. The 10,000 × g supernatant contains membrane protein.

### Statistical analysis

Paired or non-paired student $t$-test was used to determine statistical significance, defined at p-value<0.05. * denotes $p<0.05$ by $t$-test, ** denotes $p<0.01$ by $t$-test. Unless otherwise specified, results were presented as mean ±standard error of the mean (SEM). Experiments were performed at least twice. Sample size was determined based on prior experience with similar in vivo studies.

## Acknowledgements

We are grateful to Dr. Peter Espenshade (Johns Hopkins) for the gift of the SCAP-deficient HEK293 cells. We thank Dr. Steven Bensinger, Dr. Simon Beaven, and members of the Tontonoz Laboratory and the UCLA Atherosclerosis Research Unit for useful discussions and technical support. This work was supported by NIH grants HL030568 and DK063491 (to PT) and American Heart Association Scientist Development Grant 16SDG31180008 (to LZ).

## Additional information

### Competing interests

Peter Tontonoz: Reviewing Editor, eLife. The other authors declare that no competing interests exist.

### Funding

| Funder | Grant reference number | Author |
|---|---|---|
| Howard Hughes Medical Institute | | Peter Tontonoz |
| American Heart Association | Scientist Development Grant (16SDG31180008) | Li Zhang |
| National Heart, Lung, and Blood Institute | HL088528 | Ryan Temel |
| National Heart, Lung, and Blood Institute | HL111932 | Ryan Temel |
| National Institutes of Health | HL030568 | Peter Tontonoz |
| National Institutes of Health | DK063491 | Peter Tontonoz |

The funders had no role in study design, data collection and interpretation, or the decision to submit the work for publication.

### Author contributions

Li Zhang, Peter Tontonoz, Conceptualization, Formal analysis, Supervision, Funding acquisition, Investigation, Visualization, Writing—original draft, Project administration, Writing—review and editing; Prashant Rajbhandari, Conceptualization, Validation, Investigation, Methodology, Writing—original draft, Writing—review and editing; Christina Priest, Resources, Data curation, Formal analysis, Methodology; Jaspreet Sandhu, Data curation, Formal analysis, Methodology; Xiaohui Wu, Investigation, Methodology, Writing—review and editing; Ryan Temel, Data curation, Methodology; Antonio Castrillo, Data curation, Formal analysis, Writing—review and editing; Thomas Q de Aguiar Vallim, Conceptualization, Data curation, Supervision, Funding acquisition, Investigation, Writing—review and editing; Tamer Sallam, Resources, Data curation, Methodology

**Author ORCIDs**

Peter Tontonoz (iD) http://orcid.org/0000-0003-1259-0477

**Ethics**

Animal experimentation: This study was performed in strict accordance with the recommendations in the Guide for the Care and Use of Laboratory Animals of the National Institutes of Health. All of the animals were handled according to approved institutional animal care and use committee (IACUC) protocols (#99-131) of the University of California, Los Angeles.

**Decision letter and Author response**

Decision letter https://doi.org/10.7554/eLife.28766.017
Author response https://doi.org/10.7554/eLife.28766.018

## Additional files

**Supplementary files**

• Transparent reporting form
DOI: https://doi.org/10.7554/eLife.28766.014

**Major datasets**

The following dataset was generated:

| Author(s) | Year | Dataset title | Dataset URL | Database, license, and accessibility information |
|---|---|---|---|---|
| Castrillo A | 2017 | LXRa LXRb Macrophage ChiP-seq | https://www.ncbi.nlm.nih.gov/geo/query/acc.cgi?acc=GSE104027 | Publicly available at NCBI Gene Expression Omnibus (accession no: GSE104027) |

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
