## [Decision Letter]

Thank you for submitting your article "Inhibition of Cholesterol Biosynthesis through RNF145-Dependent Ubiquitination of SCAP" for consideration by *eLife*. Your article has been reviewed by three peer reviewers, and the evaluation has been overseen by a Reviewing Editor and Mark McCarthy as the Senior Editor.

The reviewers have discussed the reviews with one another and the Reviewing Editor has drafted this decision to help you prepare a revised submission.

Summary:

Zhang and colleagues describe results from a series of studies investigating the role of the ER-resident ubiquitin E3 ligase RNF145 as an LXR target that regulates the SREBP-SCAP pathway by adding ubiquitin chains. This study identifies a new mechanistic control on cholesterol homeostasis, which is positioned at the interface of LXR and SREBP2. Furthermore, it reveals a function of a ubiquitin ligase (Rnf145) that, up until now, did not have any identified function, and demonstrates a new post-translational modification and regulatory action on a key cholesterol sensor, Scap. The resultant model is that high cholesterol triggers LXR activation, which in turn increases transcription of RNF145 and ubiquitination of SCAP at lysines 454 and 466, resulting in a decrease in COPII-mediated SREBP-2 processing. The net result is that ubiquitin modified SCAP-SREBP is retained in the ER and SREBP activation is blunted. This is an attractive mechanism, but several issues need to be addressed.

Essential revisions:

The major issue that needs to be addressed is data demonstrating that endogenous SREBP-2 is altered when RNF145 is knocked down and/or over expressed. Without this information, it would be unclear whether the proposed regulatory pathway operates under physiologic conditions.

The localization studies of RNF145, while likely correct, are insufficient. Negative controls for Golgi, mito, etc. should be shown. Also, a western blot of fractions would help to show RNF145 localization (if there is a sufficient antibody).

There is no assessment of direct protein interactions between RNF145 and SCAP. Although this might be difficult to show, did the authors try this by IP and mass spec?

The gene regulations studies with overexpression and KO of RNF145 are convincing. However, it is not clear to me why this regulation of SCAP affects only SREBP-2. Do the authors have a hypothesis? Can it be tested? SCAP regulation should affect both SREBPs.

The effects of RNF145 knockdown and over expression on SREBP-2 protein expression in mice in vivo when RNF145 levels are manipulated needs to be directly analyzed.

It is somewhat surprising that the role for SCAP in regulating SREBP-1 does not appear to be mentioned here?

Harvesting mice after a 6hr fast might not be best for measuring anabolic processes (especially SREBP driven processes). Was circadian time controlled as well at time of harvest?

It would be important to document degree of over-expression and knockdown for RNF145 WT and mutant proteins in vivo in Figures 2-4. Without this information it is hard to interpret lack of mutant function and whether an increase in RNF145 mediated effects should be expected.

The major proposed effect of the RNF145 manipulation is on SREBP-2 activity, given that results for a liver specific knockout of SREBP-2 was recently published by Horton et al. (PMC5348127). A comparison of the current results with those for the SREBP-2 loss should be included in the Discussion.

To rigorously interpret the reduced ubiquitination of SCAP mutants in Figure 3C and E, the level of immunoprecipitated HSV-SCAP needs to be included to control for variations in immunoprecipitation.

Given the previous literature on Scap and its established direct regulation by cholesterol and by Insig, it is unclear where this LXR-Rnf145-SREBP2 circuit fits into the normal physiology. In the Discussion, the authors should try to unwrap this complexity and perhaps even provide an illustrated model of all the mechanisms in play.

The authors show SREBP2 target genes are altered by Rnf145 over-expression or deficiency, in vivo. However, the most relevant end-point of this paper is cholesterol synthesis. Thus, the authors might want to consider measuring de novo synthesis of cholesterol in the livers of at least one of these mouse models.

---

## [Author Response]

Essential revisions:The major issue that needs to be addressed is data demonstrating that endogenous SREBP-2 is altered when RNF145 is knocked down and/or over expressed. Without this information, it would be unclear whether the proposed regulatory pathway operates under physiologic conditions.

We appreciate the suggestion. We agree that data concerning the regulation of endogenous SREBP-2 is important to support the conclusions of this study. In order to investigate the processing of endogenous SREBP-2 in response to altered levels of RNF145, we fractionated nuclei from liver tissue of mice overexpressing RNF145. We did not detect any change in the precursor form of SREBP-2 in the membrane fraction in response to RNF145. However, in the nuclear fraction, we found that the cleaved, mature SREBP-2 was clearly decreased in mice expressing RNF145 (new Figure 3I). These data are consistent with the decreased mRNA expression of SREBP-2 target genes in mice expressing RNF145.

The localization studies of RNF145, while likely correct, are insufficient. Negative controls for Golgi, mito, etc. should be shown. Also, a western blot of fractions would help to show RNF145 localization (if there is a sufficient antibody).

Thank you for this constructive suggestion. We have provided additional data to strengthen the conclusions about RNF145 location as follows. First, in addition to the ER marker calnexin shown in the original manuscript, we have performed immunofluorescence microscopy with another ER marker, Sec26. We found that RNF145 co-localized with Sec26, confirming its ER localization (new Figure 2). Second, we assessed colocalization with the mitochondria marker protein TOM20. We found that the signals for RNF145 and Tom20 were largely non-overlapping, further supporting our conclusions (new Figure 2).

There is no assessment of direct protein interactions between RNF145 and SCAP. Although this might be difficult to show, did the authors try this by IP and mass spec?

We have in fact tried to detect the interaction between RNF145 and SCAP with co-immunoprecipitation assays in cells overexpressing both proteins. However, we could not detect an interaction (new Figure 6—figure supplement 1A). This result is not unexpected, because it is common for the interaction between an E3 ubiquitin ligase and its substrate to be dynamic, transient, and unstable, rendering it difficult to detect in such assay. As an example, one cannot detect the interaction of IDOL with the LDLR in standard IP assays.

The gene regulations studies with overexpression and KO of RNF145 are convincing. However, it is not clear to me why this regulation of SCAP affects only SREBP-2. Do the authors have a hypothesis? Can it be tested? SCAP regulation should affect both SREBPs.

The reviewer raises an interesting issue. To address this, we performed experiments to compare the time courses of RNF145 regulation of SREBP-2 versus SREBP-1 target genes. The data indicate that both SREBP-2 and SREBP-1 target genes are rapidly inhibited by RNF145 overexpression. However, the expression levels of SREBP-1 target genes rebounded, whereas the expression of SREBP-2 target genes remained suppressed (Figure 6—figure supplement 1E). These findings suggest a compensatory response of SREBP-1 targets. We conclude that the observation that RNF145 primarily affects SREBP-2 target genes on day 6 is unlikely to be due to selective effects of SCAP on SREBP-2, but is rather attributable to differential compensatory responses. This issue has now been discussed in the manuscript. Further research will be needed to address the molecular basis for these responses.

The effects of RNF145 knockdown and over expression on SREBP-2 protein expression in mice in vivo when RNF145 levels are manipulated needs to be directly analyzed.

This same point was raised by another reviewer. Please see our first response above.

It is somewhat surprising that the role for SCAP in regulating SREBP-1 does not appear to be mentioned here?

This same point was raised by another reviewer. Please see our fourth response above.

Harvesting mice after a 6hr fast might not be best for measuring anabolic processes (especially SREBP driven processes). Was circadian time controlled as well at time of harvest?

For all studies in this paper, the circadian time, as well as the time of harvest, were strictly controlled. All mice in the animal experiments were housed under a 12-hour light (6am-6pm), 12-hour dark (6pm-6am) cycle. Mice in the experimental groups, as well as those in the respective control groups, were fasted at 9 am and harvested at 3pm.

It would be important to document degree of over-expression and knockdown for RNF145 WT and mutant proteins in vivo in Figures 2-4. Without this information it is hard to interpret lack of mutant function and whether an increase in RNF145 mediated effects should be expected.

Thank you for this comment. The expression levels of RNF145 in the knockdown experiment are reported in Figure 7A. We have now also included the expression levels of wild-type and mutant RNF145 in overexpression experiment (Figure 5—figure supplement 1).

The major proposed effect of the RNF145 manipulation is on SREBP-2 activity, given that results for a liver specific knockout of SREBP-2 was recently published by Horton et al. (PMC5348127). A comparison of the current results with those for the SREBP-2 loss should be included in the Discussion.

We appreciate the reviewer’s suggestion. We have now addressed this issue in the Discussion of this manuscript as requested.

To rigorously interpret the reduced ubiquitination of SCAP mutants in Figure 3C and E, the level of immunoprecipitated HSV-SCAP needs to be included to control for variations in immunoprecipitation.

We agree with the reviewers on this issue, and we have now included the data on immunoprecipitated HSV-SCAP in the manuscript as requested (revised Figures 6A and 6C).

Given the previous literature on Scap and its established direct regulation by cholesterol and by Insig, it is unclear where this LXR-Rnf145-SREBP2 circuit fits into the normal physiology. In the Discussion, the authors should try to unwrap this complexity and perhaps even provide an illustrated model of all the mechanisms in play.

We have now added extensive discussion on this issue in the revised manuscript. Cholesterol biosynthesis and clearance are subjected to complex regulation by multiple redundant pathways, e.g. HMCGR degradation and PCSK9 in addition to the INSIG-SCAP-SREBP2 pathway. We certainly do not dispute the established physiological role of the INSIG-SCAP-SREBP-2 pathway. Our data are consistent with the hypothesis that RNF145 is an additional modifier that may contribute in certain contexts. We have included additional discussion of this issue in the manuscript as requested.

The authors show SREBP2 target genes are altered by Rnf145 over-expression or deficiency, in vivo. However, the most relevant end-point of this paper is cholesterol synthesis. Thus, the authors might want to consider measuring de novo synthesis of cholesterol in the livers of at least one of these mouse models.

We appreciate the reviewer’s suggestion on this issue and we would like to eventually optimize a non-radioactive approach to this question. Unfortunately, it is our understanding that the levels of tritium required for the in vivo cholesterol synthesis assay described by the Dallas group are not permitted in research laboratories in the State of California.